# SELF-JOINT SUPERVISED LEARNING

**Navid Kardan**[†*]**, Mitchell Hill**[‡] **& Mubarak Shah**[†]
[†]Center for Research in Computer Vision
[†]Department of Computer Science
[‡]Department of Statistics and Data Science
University of Central Florida
[†]`{Kardan,Shah}@{knights,crcv}.ucf.edu`
[‡]`Mitchell.Hill@ucf.edu`

## ABSTRACT

Supervised learning is a fundamental framework used to train machine learning systems. A supervised learning problem is often formulated using an i.i.d. assumption that restricts model attention to a single relevant signal at a time when predicting. This contrasts with the human ability to actively use related samples as reference when making decisions. We hypothesize that the restriction to a single signal for each prediction in the standard i.i.d. framework contributes to well-known drawbacks of supervised learning: making overconfident predictions and vulnerability to overfitting, adversarial attacks, and out-of-distribution data. To address these limitations, we propose a new supervised learning paradigm called self-joint learning that generalizes the standard approach by modeling the joint conditional distribution of two observed samples, where each sample is an image and its label. Rather than assuming samples are independent, our models explicitly learn the sample-to-sample relation of conditional independence. Our framework can naturally incorporate auxiliary unlabeled data to further improve the performance. Experiments on benchmark image datasets show our method offers significant improvement over standard supervised learning in terms of accuracy, robustness against adversarial attacks, out-of-distribution detection, and overconfidence mitigation. Code: github.com/ndkn/Self-joint-Learning

## 1 INTRODUCTION

Neural networks lay at the heart of deep learning success. However, problems such as overfitting (Weigend, 1994; Hawkins, 2004) and confidence miscalibration (Guo et al., 2017) persist across the vast majority of supervised models. Model calibration requires the predictive confidence of a classifier to be faithful to the ground truth, meaning the expected accuracy of all samples with $m\%$ confidence should be almost $m\%$. However, deep models tend to produce overconfident predictions (Guo et al., 2017), which undermines utilization of their confidence values. Consequences of miscalibration such as susceptibility to adversarial attacks (Szegedy et al., 2013; Goodfellow et al., 2014; Madry et al., 2017; Akhtar & Mian, 2018; Akhtar et al., 2021; Edraki et al., 2021) and poor performance on out-of-distribution (OOD) examples (Liang et al., 2017; Hendrycks et al., 2018) limit the utilization of deep learning systems in real-world applications.

Existing tools to combat overfitting and miscalibration include artificially increasing training data through data augmentation (Shorten & Khoshgoftaar, 2019), using loss penalties, utilizing stochasticity to constrain a model's ability to learn data idiosyncrasies (Larsen & Hansen, 1994; Wan et al., 2013; Gal & Ghahramani, 2016), and smoothing labels to moderate overconfident predictions (Szegedy et al., 2016; Müller et al., 2019). While increasing the number of samples during training using data augmentation has been explored, an unexplored direction is to increase the number of samples used when making a prediction. Even a well-trained model could benefit from having additional data available at the time of inference, similar to a human expert that can still benefit from having a selection of reference examples when making a decision about a single test example. We hypothesize the problems of overfitting and miscalibration could be further exacerbated by limitations of the commonly used model $p(Y|X;\theta)$ for a single sample-label pair $(X,Y)$ and parameter $\theta$.

---

[*]Corresponding author

This model form only allows other training pairs $(X', Y')$ to provide reference information for the conditional probability $p(Y|X; \theta)$, and therefore the model prediction for $X$, implicitly via training and not at all during inference.

In this paper, we propose a new paradigm called self-joint supervision to construct models that make predictions using a given test signal and a randomly selected sample of reference signals (typically a subset of training or validation pairs). This work focuses on image signals but the framework applies broadly. During training, a self-joint model learns to predict the joint distribution of a pair of labels given their associated images. By training with a pair instead of one sample, the model explicitly learns sample-to-sample relationships. In this work, we retain the standard i.i.d. assumption of supervised learning and our models actively learn the relationship of pairwise conditional independence (see Appendix A.1 for a brief review of these concepts). *Learning conditional independence is the key innovation of our work and the main difference from standard supervised learning, which has no explicit cross-sample modeling.*

We apply Maximum Likelihood to learn self-joint models in the similar framework as a standard model. After training, the model infers individual labels by marginalization of the joint distribution output. We propose inference procedures to produce both a stochastic and a deterministic output. Figure 1 shows an intuitive visualization of the self-joint training and inference procedures. Experiments with stochastic self-joint models provide evidence that learning conditional independence among sample pairs can lead to models more resistant to adversarial attacks and OOD data. Moreover, our method can naturally be extended to incorporate auxiliary unannotated data during training by using marginal prediction to generate pseudo-labels. Our experiments suggest adding partially annotated data can further safeguard against OOD data.

Overall, our findings provide a novel approach to train more consistent and robust deep models and an alternative strategy to utilize auxiliary data. The main contributions of our work are summarized below.

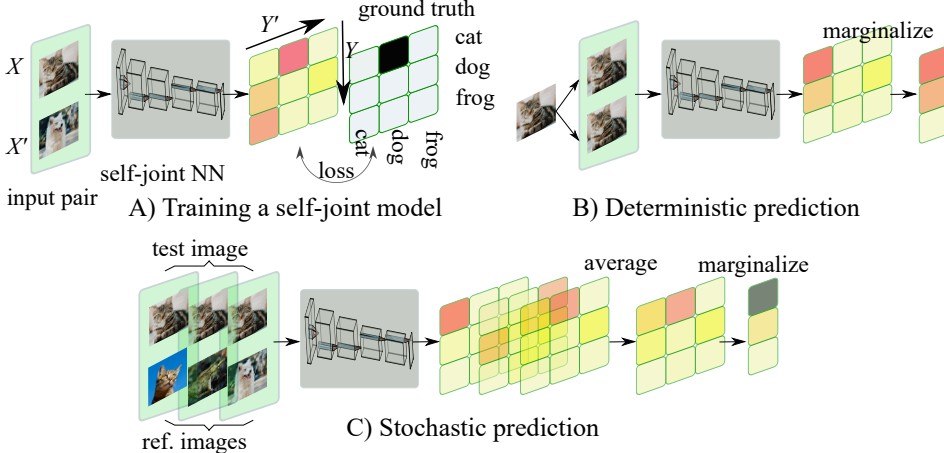

Figure 1: Self-joint paradigm effectively expands a $c$-class classification problem into a $c^2$-class classification problem. As a concrete example, assume we are facing a ternary $cat$ vs. $dog$ vs. $frog$ classification problem. Self-joint framework principally casts this into a new 9-way classification problem $\{(cat, cat), (cat, dog), ..., (frog, dog), (frog, frog)\}$, which represents all possible pairs of three labels for two inputs (in our implementation we concatenate inputs channel-wise) as shown by the $3 \times 3$ matrix above. During inference, one can extract output probabilities for each input separately through marginalization, i.e. summing up the rows or columns of the matrix. Intuitively, if the marginal prediction for a given test sample changes when we replace the second sample in the pair, this indicates the marginal predictions have weak confidence. However, if the model is consistent with its prediction across paired samples, the overall marginal prediction has high confidence. This property leads to increased robustness.

(1) We introduce a novel perspective of self-joint learning where the model can accommodate sample dependency. By applying the model to pairs of i.i.d. data (pairs are i.i.d.), we learn rather than assume conditional independence between labels given two input images. (2) We show self-joint

learning leads to substantial improvement in classification accuracy as the network capacity scales to sizes that were previously found to be detrimental due to overfitting. (3) We construct a new type of stochastic models that can provide robust predictions for adversarial examples and detect OOD data. In particular, we explore the impact of our approach on model robustness against adversarial attacks and show it drastically improves adversarial robustness against small perturbations. (4) We propose a novel technique to inject priors into learning models with the help of auxiliary unlabelled data. Our experiments show that utilizing auxiliary data can improve the model's ability to recognize OOD data.

## 2 RELATED WORK

**Out-of-Distribution (OOD) Detection.** OOD data play a vital role in real-world applications, where the model commonly encounters samples from novel classes. Ideally, the model should produce low confidence predictions with high level of uncertainty for data from unknown classes. However, several studies suggest that deep models frequently make confident predictions for OOD data (Kardan & Stanley, 2016; 2017; DeVries & Taylor, 2018). It is now an active area of research to mitigate this drawback (Li & Gal, 2017; DeVries & Taylor, 2018; Ren et al., 2019; Chen et al., 2020a; Malinin & Gales, 2018; Kardan et al., 2021). In recent studies, the majority of dedicated techniques for OOD detection assume access to part of OOD data (Liang et al., 2017), which makes OOD detection a new binary classification problem. However, in practice, this is an unrealistic assumption. In a different approach, outlier exposure (Hendrycks et al., 2018) advocates applying different distributions of OOD data for train and test and achieve substantial improvement over standard baseline. In this work we apply the same setup for evaluation of our OOD experiments with auxiliary data.

**Adversarial Examples.** Adversarial examples, which use a deliberate minute perturbation of input samples that translates to significant modification of model's prediction, are yet another shortcoming that complicates adoption of deep models in security-critical applications such as face recognition and autonomous driving vehicle systems. In addition, these samples point out an intrinsic difference between how inference occurs in humans and current deep models. Crafting a defense against adversarial attacks is an active area of research that witnessed a myriad of algorithms that were subsequently broken by more advanced attacks. A significant exception is adversarial training (Madry et al., 2017; Kannan et al., 2018; Zhang et al., 2019), where the deep model is exposed to adversarial examples during the course of training to induce a degree of immunity against them. However, adversarially trained models suffer from performance degradation on clean (unperturbed) samples. Our proposed framework preserves classification performance on clean images, in addition to improving performance against adversarial examples by performing a robust prediction.

**Stochastic Neural Networks.** Several studies show that applying stochastic models, such as Bayesian neural networks (MacKay, 1992; Barber & Bishop, 1998; Hinton & van Camp, 1993; Hernández-Lobato & Adams, 2015; Gal & Ghahramani, 2016), can mitigate overfitting and improve resistance against OOD and adversarial data. Yet, the main challenge with these approaches is scaling these models up to deep models with millions of parameters. Another technique to construct stochastic deep neural networks is Monte-Carlo dropout (MC-dropout) proposed by Gal & Ghahramani (2016). This approach applies dropout (Srivastava et al., 2014) during both train and inference time. The Monte-Carlo sampling at inference time can produce reliable uncertainty estimations, therefore, MC-dropout has been successfully applied to detect OOD Li & Gal (2017) and adversarial examples Feinman et al. (2017); Rawat et al. (2017); Li & Gal (2017). Similarly, self-joint models produce reliable predictions by performing a Monte-Carlo sampling during inference.

**Mutual Information for Representation Learning.** Paired data has been widely applied for representation learning by mutual information (Becker & Hinton, 1992; Dhillon et al., 2003; Wang et al., 2010; Friedman et al., 2013). In particular, paired data has been previously applied for co-clustering (Dhillon et al., 2003; Wang et al., 2010). In deep networks, IMSAT (Hu et al., 2017), DeepINFOMAX (Hjelm et al., 2018), and ICC (Ji et al., 2019) pursue better features by maximizing information between different forms of representation of a pair of variations of the same sample. Although these approaches learn from a pair of images, unlike self-joint paradigm, they do not learn a joint distribution between the two samples in the pair; They merely apply the same feature extractor to each sample. Furthermore, these approaches maximize mutual information between different instances (augmentations) of the same sample.

**Contrastive Learning for Representation Learning.** There is a growing body of research, such as CPC (Henaff, 2020) and SimCLRv2 (Chen et al., 2020b), that tries to learn representations from the

similarity between a pair of samples based on the contrastive loss (Tian et al., 2019; Li et al., 2020; Chen et al., 2020b). These representations are later applied to a smaller set of samples for supervised fine-tuning. Given the abundance of unannotated data, recent studies (Goyal et al., 2021) show that this approach can surpass performance of supervised methods that have only access to a limited annotated dataset. Likewise contrastive learning, our framework learns better features based on a pair of samples. However, we learn an explicit joint distribution that can deal with both annotated and unannotated data, simultaneously, and can lead to more robust classifiers.

## 3   Self-Joint Learning Framework

This section introduces our novel approach. We focus on the task of image classification, although our method applies broadly beyond image data and the classification task. We propose to learn a stochastic classifier by training a model on the joint distribution of pairs of data samples. We leverage this framework to produce both robust stochastic predictions and fast deterministic predictions, and to incorporate auxiliary unlabelled data.

Throughout the paper, let $(X_i, Y_i)$ be a paired image random variable $X_i$ and class (label) random variable $Y_i$, and $\{(X_i, Y_i)\}_{i=1}^n$ be the training dataset. We consider the case where the state space of the predictors $X_i$ is a subset of $\mathbb{R}^m$ and the state space of responses $Y_i$ is a subset of $\mathbb{R}^c$, where $m$ is the number of image pixels times the number of color channels and $c$ is the number of classes. At times we overload the notation of $Y_i$ and treat it as a member of the set $\{1, \ldots, c\}$. Finally, we assume there is a true but unknown density $q(x, y)$ such that $(X_i, Y_i)$ pairs are i.i.d. samples from $q(x, y)$ for $i = 1, \ldots, n$. We emphasize that in this work we always assume that the observations $\{(X_i, Y_i)\}_{i=1}^n$ are truly i.i.d. from $q$, even when we later introduce a model that can accommodate sample dependency.

### 3.1   Revisiting Supervised Learning

The most common approach in supervised learning is to define a conditional density $p(Y|X; \theta)$ for a single pair $(X, Y)$ and use the Maximum Likelihood Estimator (MLE)

$$\hat{\theta}_{MLE}(\{X_i, Y_i\}_{i=1}^n) = \arg\max_{\theta} p(Y_1, ..., Y_n | X_1, ..., X_n; \theta) \tag{1}$$

$$= \arg\max_{\theta} \prod_{i=1}^n p(Y_i | X_i; \theta) = \arg\max_{\theta} \sum_{i=1}^n \log p(Y_i | X_i; \theta). \tag{2}$$

This framework covers the most widely-used loss functions and tasks such as classification with cross-entropy and regression with square-error. The transition from (1) to (2) is justified not as a mathematical consequence but as a definition

$$p(Y_1, \ldots, Y_n | X_1, \ldots, X_n; \theta) := \prod_{i=1}^n p(Y_i | X_i; \theta) \tag{3}$$

chosen by the modeler based on the i.i.d. assumption about the dataset $\{(X_i, Y_i)\}_{i=1}^n$. We note that this choice is not strictly necessary and that the modeler could use alternate factorizations. In the extreme case, the modeler could simply not factorize. Instead, the modeler could define and attempt learning the fully joint conditional distribution $p(Y_1, \ldots, Y_n | X_1, \ldots, X_n; \theta)$ directly, which is a valid but impractical approach even if the modeler still believes the data is i.i.d. Such a model would be difficult to learn, difficult to use, and unlikely to generalize well. However, the possibility of an intermediate factorization between i.i.d. and fully joint learning is an intriguing and unexplored direction that is the main focus of this work.

The assumption (3) is motivated in part by the desire to learn the conditional distribution $p(Y_i | X_i; \theta)$ for a single pair $(X_i, Y_i)$. In this work, we instead use a conditional distribution $p(Y_i, Y_j | X_i, X_j; \theta)$ for two pairs $(X_i, Y_i)$ and $(X_j, Y_j)$ as our base model. We call this the self-joint model. Although our base model could learn conditional dependency between $Y_i$ and $Y_j$ given $X_i$ and $X_j$, we will exclusively train $p(Y_i, Y_j | X_i, X_j; \theta)$ using conditionally independent samples $Y_i | (X_i, X_j)$ and $Y_j | (X_j, X_j)$. We note that this conditional independence relation trivially holds under the i.i.d. assumption. In the context of classification, conditional independence means that the $c \times c$ prediction matrix $h(X_i, X_j; \theta)$ defined by the ideal classifier $p(Y_i, Y_j | X_i, X_j; \theta)$ should have the form $h(X_i, X_j; \theta) = q_i q_j^{\mathsf{T}}$, where $q_i, q_j \in \mathbb{R}^c$ represent the true conditional distribution $q(Y_i | X_i)$ and

$q(Y_j|X_j)$, respectively (see Section 3.2 for full details). An essential difference between self-joint learning and standard supervised learning is that our model can explicitly learn the sample-to-sample relation of pairwise conditional independence, instead of assuming this via the factorization in (3).

We can once again use Maximum Likelihood to learn our self-joint model by defining

$$\hat{\theta}_{SJ}(\{X_i, Y_i\}_{i=1}^n) = \arg\max_\theta \ p(Y_1, ..., Y_n | X_1, ..., X_n; \theta) \tag{4}$$

$$= \arg\max_\theta \ \sum_{S \in \mathcal{S}} \log \prod_{(i,j) \in S} p(Y_i, Y_j | X_i, X_j; \theta) \tag{5}$$

$$= \arg\max_\theta \ \sum_{S \in \mathcal{S}} \sum_{(i,j) \in S} \log p(Y_i, Y_j | X_i, X_j; \theta), \tag{6}$$

where $S$ is a set of $\frac{n}{2}$ arbitrary non-intersecting parings $\{i, j\}$ for $i, j \in \{1, \ldots, n\}$ and $\mathcal{S}$ is the set of all such sets. This has the mild requirement that the number of samples $n$ is even. For a given $S$, the factorization in (5) is again justified by the i.i.d. assumption across all observations, and we simply choose to cease factoring at pairs of two samples. Since all such factorizations are valid under the i.i.d. assumption, we choose to average the log likelihood over all possible factorizations when defining our learning objective, which under argmax is equivalent to summing. The ideal model will learn that $\prod_{(i,j) \in S} p(Y_i, Y_j | X_i, X_j; \theta)$ remains constant across all $S$. In practice, we use a single $S$ sampled uniformly from $\mathcal{S}$ and a random subset $S_b$ of $b$ pairs in $S$ to obtain a stochastic gradient

$$\hat{\Delta}_{SJ}(\theta) = \sum_{(i,j) \in S_b} \frac{\partial}{\partial \theta} \log p(Y_i, Y_j | X_i, X_j; \theta). \tag{7}$$

This gradient is unbiased for the gradient of the objective (6) as discussed in Appendix A.2.

### 3.2 Learning and Predicting with Self-Joint Classifiers

In this section we will apply the self-joint supervision method to the image classification task. Let $p(Y_i, Y_j | X_i, X_j; \theta) := h(X_i, X_j; \theta)$ be a classifier that maps two samples $X_i, X_j \in \mathbb{R}^m$ to a prediction for their corresponding joint label $Y_i Y_j^\mathsf{T} \in \mathbb{R}^{c \times c}$, where $Y_i, Y_j \in \mathbb{R}^c$ are the labels of $X_i$ and $X_j$, respectively. Since $Y_i$ is a probability density, $||Y_i||_1 = 1$ and $0 \leq Y_{i,k} \leq 1$ for each scalar component $Y_{i,k}$ of $Y_i$, and the same applies to $Y_j$. It is straightforward to show that the total loss

$$L(\theta; \{X_i, Y_i\}_{i=1}^n) = \sum_{S \in \mathcal{S}} \sum_{(i,j) \in S} l(\theta; X_i, X_j, Y_i, Y_j), \tag{8}$$

$$\text{where} \quad l(\theta; X_i, X_j, Y_i, Y_j) = (Y_i Y_j^\mathsf{T}) \cdot \log h(X_i, X_j; \theta) \tag{9}$$

with the dot product taken in the sense of vectors and log taken elementwise, is equivalent to the MLE learning objective in (6) for cross-entropy loss, and that the stochastic gradient can optimize this loss similar to (7). See Appendix A.3 for details of this equivalence between (6) and (8).

Although we are learning a joint model $p(Y, Y'|X, X'; \theta)$, we still would like to make predictions for a single test image $X$ via $p(Y|X; \theta)$ as in the standard i.i.d. framework. Note $X$ and $X'$ denote the first and second inputs of the model, respectively, and $Y$ and $Y'$ are their corresponding labels. This can naturally be accomplished by marginalization and Monte Carlo approximation:

$$p(Y|X; \theta) = \int_{\mathbb{R}^m} \sum_{k=1}^c p(Y, Y' = k | X, X'; \theta) q(X') dX' \tag{10}$$

$$\approx \frac{1}{|B|} \sum_{i \in B} \sum_{k=1}^c p(Y, Y' = k | X, X_i; \theta) = \hat{p}(Y|X; \theta), \tag{11}$$

where $B$ denotes a random subset of $\{1, \ldots, n\}$ with $|B|$ giving its cardinality, and $q(x)$ is the marginal of the true joint distribution $q(x, y)$. Our model can only approximate $p(Y|X; \theta)$ instead of having a closed form solution because different samples $B$ yield different predictors, but this estimate will be stable for large $|B|$ due to the weak law of large numbers. In the case where $|B|$ is moderately large, we empirically observe that predictive uncertainty estimated by either the variance of $\hat{p}(Y|X)$ or its entropy is more robust than the uncertainty calibration of a standard i.i.d. supervised

model, which provides a compelling motivation for exploring the generalization properties of our framework.

Note that (11) has a computational complexity of $O(|B|)$ and that $\hat{p}(Y|X;\theta)$ requires additional samples $X_i$ for estimation, which results in stochasticity. In our experiments, $|B|$ is typically less than or equal to 50, which is a reasonable but non-trivial additional cost. To address these concerns, we note that we can use the same $X$ for both input signals, yielding the deterministic classifier

$$\tilde{p}(Y|X;\theta) = \sum_{k=1}^{c} p(Y, Y' = k|X, X; \theta). \tag{12}$$

This is a valid estimate for (10) with $|B| = 1$ provided that $X$ is in-distribution.

Intuitively, in the Monte-Carlo inference in (11), the joint label is affected by the probabilistic label of both input samples. Generally, the two samples produce different levels of confidence and might belong to different classes, therefore, for a better approximation, we repeat our estimation of the prediction for the test sample several times. However, in (12) we know that the two sample inputs have the same label and seeing a replicate of a sample input does not add any new information about its label. Empirically, we observed that (12) often produces similar classification accuracy to (11).

### 3.3 INCORPORATING AUXILIARY DATA

Interestingly, conditional independency provides a principled tool to incorporate auxiliary data in the self-joint learning framework. Recall that semi-supervised learning (Chapelle et al., 2009; Grand-valet et al., 2005; Lee et al., 2013), self-supervised learning (Dosovitskiy et al., 2014; Oord et al., 2018), and OOD detection frameworks also typically take advantage of auxiliary data. Similarly, self-joint framework can benefit from un-labelled data. To this end, we can learn models such that the predictions on training data remain conditionally independent given an auxiliary unlabelled image. Following this approach, we effectively train self-joint models that can handle OOD data.

To incorporate auxiliary unlabelled data, we still assume that all data are generated in i.i.d. pairs $(X_i, Y_i)$, but for the auxiliary data we have unobserved labels which we denote by $Y_i = \emptyset$. The factorization in (6) still holds under this assumption, but we are unable to learn the model without a numerical value if $Y_i = \emptyset$. A straightforward estimate (similar to pseudo-labels to replace one-hot encoding) is

$$\hat{Y}_i = \sum_{k=1}^{c} p(Y_i, Y_j = k|X_i, X_j; \theta), \tag{13}$$

which is the marginalization of the predictor matrix $h(X_i, X_j; \theta)$. The same principle applies for missing $Y_j$ by summing over the rows. This approach can even be used when both $Y_i = \emptyset$ and $Y_j = \emptyset$. When learning with auxiliary data, we replace any null $Y_i$ with the pseudo-label $\hat{Y}_i$ in the objective function (6). Intuitively, auxiliary samples direct a self-joint model to preserve its predictions when it encounters such auxiliary samples. With the right choice of auxiliary data, it can improve generalization of the model. For example, if auxiliary data share similar background as original data, the model learns to ignore those superficial features and focus on salient foreground characteristics of the samples.

## 4 EXPERIMENTS

In this section, we evaluate the proposed framework on four visual classification tasks, i.e., CIFAR-10, CIFAR-100, SVHN, and STL-10. The generalized self-joint training procedure is summarized in Algorithm 1 and we use this framework across all experiments. To make general conclusions, we refrain from hyperparameter optimization for any specific dataset and apply almost the same configuration (data augmentation, training recipe, and model architecture) for all datasets (details in Appendix A.5). The overall results provide strong evidence that the new framework dramatically improves classification performance, adversarial and OOD data robustness, and facilitates training of large neural networks.

### 4.1 REGULARIZATION EFFECT OF SELF-JOINT FRAMEWORK

To demonstrate the regularization effect of our framework, we trained several wideResNet (Zagoruyko & Komodakis, 2016; He et al., 2016) models with different capacities on CIFAR-10 dataset in standard and self-joint frameworks. In this experiment, we increase the capacity by fixing

---

**Algorithm 1** Training procedure for self-joint learning framework

---

**Input** Training set $\mathcal{D}_{\text{train}} = \{(X_i, Y_i)\}_{i=1}^n$, auxiliary set $\mathcal{D}_{\text{aux}} = \{(X_j, \emptyset)\}_{j=1}^{n_{\text{aux}}}$, loss function $l$,
model $p(Y, Y'|X, X'; \theta) = h(X, X'; \theta)$, learning rate $\alpha$, number of classes $c$,
batch size $|\mathcal{B}|$

1. **set** $\mathcal{D} \leftarrow \mathcal{D}_{\text{train}} \cup \mathcal{D}_{\text{aux}}$
2. **repeat**
3.     $\mathcal{B} \leftarrow \{(X_i, Y_i)\}$ for $i \sim \{1, ..., |\mathcal{D}|\}$         ▷ construct batch set
4.     $L \leftarrow 0$         ▷ loss
5.     **for** $i = 1$ **to** $|\mathcal{B}|$ **step size** 2
6.         $j \leftarrow i + 1$
7.         **if** $Y_i = \emptyset$
8.             $Y_i \leftarrow \sum_{k=1}^c [h(X_i, X_j; \theta)]_{k,:}$     ▷ Eq.13: marginalizing out the output space for $X_i$
9.         **if** $Y_j = \emptyset$
10.            $Y_j \leftarrow \sum_{k=1}^c [h(X_i, X_j; \theta)]_{:,k}$     ▷ Eq.13: marginalizing out the output space for $X_j$
11.         $L \leftarrow L + l(h(X_i, X_j; \theta), Y_i Y_j^{\mathsf{T}})$     ▷ Eq.8: updating loss
12.     $\theta \leftarrow \theta + \alpha \nabla_\theta L$
13. **until convergence**

---

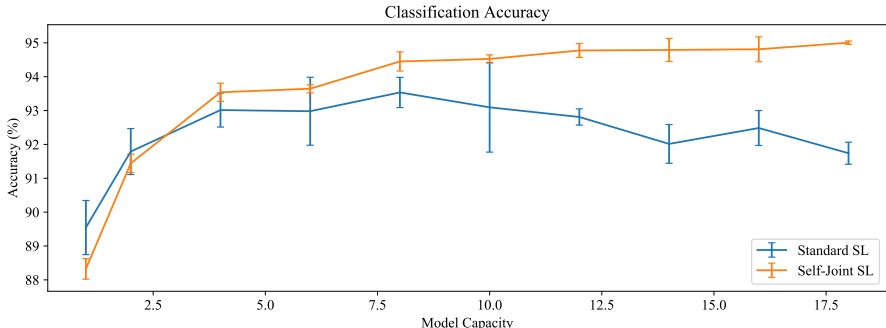

Figure 2: Visualization of overfitting effect for similar models trained in standard and self-joint framework on CIFAR-10. Standard supervised models with high capacity, measured on the $x$-axis using the width in standard WideResNet-34, suffer from overfitting as shown by the drop in accuracy of the blue line around 10. Self-joint learning can successfully train larger networks due to its regularization effect.

the depth (34 layers) and increasing the width (number of channels) of the models. All the models utilize the same training regime (hyperparameters, augmentations, etc) for training. Figure 2 shows the different classification accuracies for models with different levels of capacity in each framework. In this plot, due to overfitting, increasing capacity after a certain level harms the model performance in the standard supervised learning framework. However, self-joint framework, due to its strong regularization effect, is able to achieve increasingly higher levels of accuracy with higher capacity models.

## 4.2 MODEL ACCURACY AND CALIBRATION EXPERIMENTS

In this experiment, we compare trained models with similar architecture in the standard and self-joint frameworks. We train the models in full and low training-sample regimes, where in the latter we reduce training data to $1\%$ of the total data. Table 1 depicts the classification accuracy for different models with these two setups on CIFAR-10/100, SVHN, and STL-10. The results suggest that proposed method can effectively train large models that are able to achieve higher accuracy than standard supervised learning models in both experiments. This indicates strong regularization (Goodfellow et al., 2016) effect of the proposed framework.

Following common practice in model calibration, we report (Table 1) negative log-likelihood (NLL) as an indicator of the quality of predictive confidences. For more detail of this metric refer to (Guo et al., 2017). Unlike common practice in model calibration literature, we do not run any post-processing step to calibrate models, but rather measure the raw model confidence values to compare how different techniques affect predictive confidence. In this experiment it is evident that self-joint models outperform standard models in model calibration. Moreover, interestingly, we observe while

Table 1: Classification accuracy (%) and model calibration measured by negative-log-likelihood (NLL) on different datasets. Higher accuracy and lower NLL are better.

|  | Model | CIFAR-10 | CIFAR-100 | STL-10 | SVHN |
|---|---|---|---|---|---|
| Acc. with 1% training data | Standard | 42.50±1.93 | 6.67±0.97 | 22.1±2.95 | 82.48±2.32 |
|  | SJ (ours) | **55.52±0.18** | **12.45±0.16** | 26.06±1.9 | **88.98±0.80** |
| Acc. with full training data | Standard | 95.68±0.03 | 79.14±0.11 | 73.95±1.08 | 97.48±0.07 |
|  | SJ (ours) | **97.03±0.04** | **81.16±0.03** | **87.69±0.51** | 97.59±0.1 |
| Model cal. on full training data | Standard | 0.26±0.02 | 1.09±0.03 | 1.56±0.05 | 0.15±0.03 |
|  | SJ (ours) | 0.22±0.01 | **0.85±0.02** | **0.55±0.06** | 0.15±0.0 |

Table 2: Adversarial robustness accuracy(%) of different models measured by the classification accuracy against adversarial examples generated by PGD with different levels of perturbation ($\epsilon$).

|  | CIFAR-10 | CIFAR-100 | STL-10 | SVHN | CIFAR-10 | CIFAR-100 | STL-10 | SVHN |
|---|---|---|---|---|---|---|---|---|
| Model |  | $\epsilon = 2/256$ |  |  |  | $\epsilon = 4/256$ |  |  |
| Standard | 33.2 | 9.52 | 21.22 | 63.04 | 22.06 | 5.85 | 14.0 | 46.33 |
| Dropout | 44.91 | 14.6 | 34.1 | 67.47 | 33.42 | 5.25 | 18.08 | 46.08 |
| SJ (ours) | **62.29** | **32.65** | **48.12** | **82.0** | **45.52** | **20.33** | **35.29** | **70.85** |

conventional supervised learning generally produce overconfident models, self-joint models tend to produce under-confident predictions. We attribute this phenomenon to the capacity of the trained models, where increasing model depth usually enhances predictive confidence.

## 4.3 ADVERSARIAL ROBUSTNESS EXPERIMENTS

This section studies the adversarial robustness of self-joint models. Some early studies have proposed MC-dropout as a defense against adversarial attacks (Feinman et al., 2017; Rawat et al., 2017; Li & Gal, 2017). Although this defense has later been broken, its positive effect on adversarial robustness has been approved in (Carlini & Wagner, 2017). Therefore, we compare against MC-dropout models as our baseline. In these experiments we apply the gold standard of first-order attacks, PGD, in a white-box setting. To compute the gradients of stochastic model, we apply expectation over transformation (Athalye et al., 2018). In this method, given a stochastic classifier $f(x; \rho)$, its gradient is calculated by: $\nabla_X E_\rho[f(X; \rho)] = E_\rho[\nabla_X f(X; \rho)] \approx \frac{1}{n} \sum_{i=1}^{n} \nabla_x f(X; \rho_i)$, where $\rho$ denotes model randomness associated with drawing a sample $\{X_{i_1}, X_{i_2}, \ldots, X_{i_{|B|}}\}$ according to the random indices $\{i_1, \ldots, i_{|B|}\} \subset \{1, \ldots, n\}$ to get the estimate $\hat{p}(Y|X; \theta)$ in (11).

Table 2 shows the average performance over several runs of different models on 500 adversarial examples that are generated from random test samples with different levels of perturbation (more results reported in Appendix A.5). We set the step size and the maximum number of iterations of PGD to $1/256$ and $4$, respectively. The stochastic models tend to outperform the rest of models for different levels of perturbation on every dataset. Nevertheless, it is clearly evident that the stochastic self-joint models (average of 20 samples) are a notch above their stochastic dropout counterparts in adversarial robustness, e.g. about $15\%$ improvement on CIFAR-100, and consistently outperform all the other baselines.

## 4.4 DETECTING OOD DATA

Detecting OOD data is another important aspect of robustness. We apply two evaluation metrics to measure the OOD detection rate of a model. The area under the receiver operating characteristic curve (AUROC) is widely used in OOD detection literature. The AUROC is the expected probability that an in-distribution data will have a higher detection score than an OOD data. We also report *detection error*, which is the highest detection rate possible that is achieved by choosing the optimal threshold.

In this experiment, we include another neural network trained with outlier exposure (OE) (Hendrycks et al., 2018), which is a dedicated OOD detection. The OE utilizes an auxiliary OOD dataset to improve OOD robustness of the model. It specifically tries to make classifier produce a uniform posterior on OOD samples (for more detail refer to (Hendrycks et al., 2018)). We included

Table 3: Average AUROC (%)/Detection Error (%) for different models. The OOD datasets include CIFAR-10/100, SVHN, STL-10, CelebA, and VOC.

| Model/in-distribution | CIFAR-10 | CIFAR-100 | STL10 | SVHN |
|---|---|---|---|---|
| Standard | 79.46/24.21 | 74.84/29.01 | 68.97/33.9 | 96.09/7.89 |
| Dropout | 81.99/22.06 | 74.69/29.55 | 70.29/32.53 | 96.6/6.96 |
| OE | **90.23/12.94** | **81.24/21.71** | 70.74/32.55 | **99.99/0.03** |
| SJ (ours) | 85.41/19.38 | 77.03/27.7 | **82.26/23.6** | 98.91/3.94 |
| Auxiliary+SJ (ours) | **91.34/12.26** | **81.29/21.72** | **84.83/20.49** | **99.99/0.13** |

OE as a representative of a strong OOD detection baseline and a method that incorporates additional auxiliary data. We also train models with auxiliary data as explained in section (3.3), which obtain the best performance. For more information about these auxiliary datasets refer to Appendix A.5.

The test set of SVHN, VOC-2012, CIFAR-10, CIFAR-100, CelebA, and STL-10 datasets is chosen for OOD datasets, where $500$ samples are randomly selected (five repetitions) for in- and out-of-distribution data to compare the detection rate of each model. Note that the majority of these datasets have little to no common classes, however STL-10 and CIFAR-10 share $80\%$ of their classes and thus an ideal detection rate is far below $100\%$ AUROC. We generally use maximum softmax probability (e.g. Hendrycks & Gimpel (2016)) as detection score, except for Monte-Carlo dropout models, where we apply entropy instead because we found that entropy is slightly superior in this case. For Monte-Carlo approximation we always compute average of $50$ samples.

Table 3 shows the average results for all OOD datasets (detailed results are available in Appendix A.5), where model stochasticity plays a crucial role in boosting OOD detection rate. Another important observation is the vital role of auxiliary data in improving detection rate of stochastic self-joint models. Interestingly, self-joint with auxiliary data generally outperforms outlier exposure, which is a dedicated state-of-the-art OOD detection technique.

### 4.5 IMPORTANCE OF JOINT DISTRIBUTION MODELING

We conduct a study to demonstrate the key role of joint distribution modeling in our framework. Here we train a model that jointly classifies two samples, simultaneously, but remove the joint output space. In particular, unlike self-joint model, this model performs the two predictions independently, where, the model includes $2 \times c$ output units for a $c$-classification problem. From another perspective, it is an ensemble of two classifiers with shared hidden units. During the test time, a sample is fed to both inputs simultaneously and the average of the two predictions are considered as the output. To have a fair comparison, we use the same WideResNet architecture as self-joint experiment except for the last layer that has $2 \times c$ outputs. We observe that not only this ensemble does not help the performance, but also it causes a $2\%$ and $3.5\%$ decline in the classification accuracy of CIFAR-10 and CIFAR-100 datasets, respectively; even an inferior accuracy than standard supervised models. Moreover, these model show overconfident predictions. These findings clearly demonstrate the superiority of self-joint models over mere ensembling of models with shared parameters (more experiments in Appendix A.5).

### 5 CONCLUSION

In this study, we proposed an alternative way to train and make inference by deep supervised models. Our framework directly learns sample-to-sample relation of conditional independence by modeling the joint conditional distribution of two samples. Our approach has several advantages over the standard supervised learning. First, it induces a strong regularization effect that can successfully train large deep models to the higher levels of accuracy than the standard models. Second, it can leverage extra set of unlabeled data, in conjunction with the training data, to improve the performance of the resulting model. Unlike self-supervised techniques, our approach works end-to-end, and, as a result learns features that are more suitable for the final learning task. Third, depending on the application, the trained models can infer in two modes: (1) a fast deterministic inference similar to standard models, and (2) a stochastic inference mode that performs more robust predictions. We demonstrated that this stochastic inference results in more robust predictions against adversarial and OOD samples. We hope this study will be a stepping stone towards more reliable learning models and inspires new avenues for future research.

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

# A APPENDIX

## A.1 INDEPENDENCE, CONDITIONAL INDEPENDENCE, AND I.I.D.

Two events, $A$ and $B$, are independent, denoted by $A \perp\!\!\!\perp B$, if:
$$p(A, B) = p(A)p(B). \tag{14}$$
Furthermore, assuming $p(C) > 0$, $A$ and $B$ are conditionally independent given $C$, denoted by $A \perp\!\!\!\perp B|C$, if:
$$p(A|B, C) = p(A|C). \tag{15}$$
Equivalently, Eq. 15 can be stated as:
$$p(A, B|C) = p(A|C)p(B|C). \tag{16}$$
Intuitively, two events are independent if they are not related, thus occurrence of one does not affect the other. In Eq. 15, two events $A$ and $B$ given $C$ are conditionally independent if occurrence of $B$ does not add any information about $A$ above what we know based on $C$. As a special case of this view:
$$p(A, B|C, C) = p(A, B|C). \tag{17}$$
Note that neither independence relationship entails conditional independence nor conditional independence entails independence.

The assumption that the training data are independent and identically distributed, or i.i.d., is common to most learning frameworks. A dataset $\{X_i, Y_i\}_{i=1}^n$ is called i.i.d. if $(X_i, Y_i) \sim q$ for some fixed density $q$ and the joint density $q'$ of the entire dataset satisfies
$$q'(X_1, Y_1, X_2, Y_2 \ldots, X_n, Y_n) = q(X_1, Y_1)q(X_2, Y_2) \cdots q(X_n, Y_n).$$
The relation of pairwise conditional independence $Y_i \perp\!\!\!\perp Y_j|X_i, X_j$ follows from the independence of the joint distribution of $(X_i, Y_i)$ and $(X_j, Y_j)$ for all $i, j$:
$$q(Y_i, Y_j|X_i, X_j) = \frac{q(Y_i, Y_j, X_i, X_j)}{q(X_i, X_j)} = \frac{q(Y_i, X_i)q(Y_j, X_j)}{q(X_i)q(X_j)} = q(Y_i|X_i)q(Y_j|X_j).$$
This justifies the use of the joint label $Y_iY_j^\mathsf{T}$ as the target for $p(Y_i, Y_j|X_i, X_j; \theta)$ in the self-joint framework, where we are overloading the notation and treating $p(Y_i, Y_j|X_i, X_j; \theta)$ as a matrix rather than scalar in the classification example.

## A.2 UNBIASEDNESS OF SELF-JOINT STOCHASTIC GRADIENT

In this section we show that the stochastic gradient (7) is unbiased for the gradient of the objective (6). We note
$$\frac{\partial}{\partial \theta} \sum_{S \in \mathcal{S}} \sum_{(i,j) \in S} \log p(Y_i, Y_j|X_i, X_j; \theta) \propto \frac{1}{|\mathcal{S}|} \sum_{S \in \mathcal{S}} \frac{1}{|S|} \sum_{(i,j) \in S} \frac{\partial}{\partial \theta} \log p(Y_i, Y_j|X_i, X_j; \theta)$$
$$= \mathrm{E}_{\mathcal{S}} \left[ \mathrm{E}_S \left[ \frac{\partial}{\partial \theta} \log p(Y_i, Y_j|X_i, X_j; \theta) \right] \right]$$
where the summations are expectations with respect to uniform distributions on $\mathcal{S}$ and $S$ respectively. The unbiasedness of (7) simply follows from the fact that each sample $S$ and $(i, j)$ come from the same uniform distributions, and the fact that the mean of unbiased estimators is also unbiased.

### A.3 EQUIVALENCE OF MAXIMUM LIKELIHOOD AND CROSS-ENTROPY LOSS FOR SELF-JOINT LEARNING

In this section we show that the self-joint cross-entropy loss (8) is equivalent to the Maximum Likelihood objective (6) in the case where the model $h(X, X'; \theta)$ represents the conditional density $p(Y, Y'|X, X'; \theta)$. This simply follows from observing that

$$p(Y, Y'|X, X'; \theta) = \prod_{k,k'} [h(X, X'; \theta)]_{k,k'}^{1\{Y_k=1, Y'_{k'}=1\}}$$

which implies

$$\log p(Y, Y'|X, X'; \theta) = Y(Y')^t \cdot \log h(X, X'; \theta).$$

### A.4 RELATIONSHIP TO BAYESIAN NEURAL NETWORKS

A successful approach to estimate uncertainty is Bayesian neural network, where every parameter of the model has a distribution (MacKay, 1992; Barber & Bishop, 1998; Hinton & van Camp, 1993; Hernández-Lobato & Adams, 2015; Gal & Ghahramani, 2016). From a different perspective, these models can be thought as an ensemble of neural networks with shared parameters. Using maximum a posteriori method, these neural networks can readily produce uncertainty estimations. Furthermore, their principled training usually improves the overall performance of the model.

Despite these desirable properties, the application of Bayesian neural networks is very limited in real-world applications. This is mainly due to their excessive computational cost, both during inference and training phases, and their inability to appropriately scale up to large amount of data. Moreover, from a technical point of view, choice of the right prior distribution in these networks is not always clear. An important exception is MC-dropout (Gal & Ghahramani, 2016), which applies dropout to construct an ensemble of neural networks that approximate a Bayesian neural network and can be readily applied to almost any conventional deep neural network.

Much like a Bayesian neural network, a self-joint neural network can be regarded as an ensemble of neural networks with shared parameters. In this point of view, every reference sample *implicitly* generates one instance of parameters of the model, i.e. a neural network. In other words,

$$p_Z(Y|X) = \int h(Z, X) dY', \tag{18}$$

where $Z$ is drawn from the marginal data distribution $q(x)$. By fixing the second input, $X$ in Eq. 18, of the self-joint model, $h$, and changing the first input, $Z$, Eq.18 is in effect generating an output distribution for $p(Y|X)$, as well as generating a distribution over every parameter in the neural network.

From this perspective, self-joint framework introduces a different viable solution for producing scalable Bayesian neural networks.

### A.5 ADDITIONAL EXPERIMENTS

This section includes the experimental setup for our experiments, the comparison between performance of deterministic and Monte Carlo inference procedures for different models, measuring the independence error of self-joint models, additional adversarial robustness experiments, and the OOD detection performance for individual OOD datasets.

#### A.5.1 EXPERIMENTAL SETUP

We apply four common visual classification datasets—i.e. CIFAR-10, CIFAR-100, STL-10, and SVHN— to train our models. Table 4 shows the characteristics of these datasets, along with their corresponding auxiliary dataset in our experiments. The auxiliary set for STL-10 dataset includes a subset of $5,000$ samples from the extra set of annotated images provided with this dataset. To make more general conclusions, we adhere to a unified experimental setting where *we try to apply the same set of hyperparameters to all four datasets.* In particular, we limit our augmentations to padding, random-Crop, and random horizontal-Flipping (except for SVHN), where an image is first

padded with four (12 for STL-10 to compensate for the higher image resolution of this dataset) pixels on every side. Then it is randomly cropped to the original size, and finally randomly horizontally flipped.

We implemented our code in Pytorch (Paszke et al., 2019). All datasets are normalized according to the mean and standard-deviation of their training set. All experiments apply WideResNet (with no trainable parameters for BatchNorm layers) architecture, where models trained on CIFAR-10/100 and SVHN (all datasets with $32 \times 32$ images) are always a $64$ layer WideResNet with width=10. The STL-10 model utilizes a deeper $124$ layer WideResNet with width$= 5$. Moreover, due to higher resolution of STL-10 images, we replaced the first convolutional layer in WideResNet with 5 kernels (stride$= 2$), and the final average pooling was replaced with an adaptive average pooling layer. These changes are made to make the STL-10 model as close as possible to other models. Thus, we did not attempt to optimize the architecture for the STL-10 dataset. The networks make no distinction between two permutations of an input pair, i.e., the input pairs $X_i, X_j$ and $X_j, X_i$ are assigned to the same class. We conduct experiments with shared parameters in the first convolutional layer for the two inputs on CIFAR-10. However, it does not lead to significant performance improvement. As a result, in our final experiments we do not share the parameters of the first convolutional layer. The learning rate always starts with $0.4$ and it decays after every 10 epochs with a constant scale of $0.81$. Note that we define an epoch to be roughly $390$, $390$, $196$, and $572$ iterations on CIFAR-10, CIFAR-100, STL-10, and SVHN, respectively. The batch size starts at $128$ ($256$ for STL-10) and is doubled every 20 epochs. Maximum number of epochs is always $180$, except for the regularization experiment, after which we did not observe any noticeable improvement in training loss in any experiment. We apply early stopping, where we discontinue training if neither the validation accuracy nor its loss is improved for more than 20 epochs. We always keep $10\%$ of test data for validation set and report the final test accuracy on the rest. The optimizer is always SGD with Nestrov (momentum=0.9, and weightDecay=$e^{-4}$). Finally, we apply exponential moving average normalization (Cai et al., 2021; Tarvainen & Valpola, 2017; Izmailov et al., 2018), where momentum$= 0.95$ and the average model is updated after every 100 iterations.

### A.5.2 Monte Carlo Inference in self-joint Framework

The robust self-joint model is primarily based on the Monte Carlo inference. Table 5 reports the performance of the trained models using the deterministic inference procedure in Eq. 12 versus the Monte Carlo estimation in Eq. 11. Overall, the deterministic inference performs well. However, Monte Carlo inference generally improves with increasing the sample size. Table 6 shows the effect of the sample size on performance of self-joint and Dropout models in Monte Carlo estimation. The results are averaged over two runs of each experiment.

### A.5.3 Measuring Independence Experiments

Recall that self-joint modeling provides a flexible framework that learns sample independency instead of assuming it. To investigate if self-joint models are able to successfully learn conditional sample-independency relations we conducted an experiment. This experiment utilizes the test samples from different datasets and measure the sample-independency for one pair according to the following independence error:

$$\mathrm{E}_{X,X'}||\tilde{p}(Y,Y'|X,X';\theta) - \sum_{k=1}^{c}\tilde{p}(Y,Y'=k|X,X';\theta)[\sum_{k=1}^{c}\tilde{p}(Y=k,Y'|X,X';\theta)]^T||_2.$$

Table 4: Overview of the datasets we use in our experiments. Note that we always keep the first $10\%$ of test data from each class as validation set and report the test accuracy on the rest.

| Dataset | Dimensionality | No. of Classes | Training Set | Test Set | Auxiliary Set |
|---------|----------------|----------------|--------------|----------|---------------|
| CIFAR-10 | 3,072 | 10 | 50,000 | 10,000 | CIFAR-100 |
| CIFAR-100 | 3,072 | 100 | 50,000 | 10,000 | CIFAR-10 |
| STL-10 | 2,654,208 | 10 | 5,000 | 8,000 | STL-10 extra |
| SVHN | 3,072 | 10 | 73,257 | 26,032 | CIFAR-100 |

Table 5: Classification accuracy of different models using deterministic versus Monte Carlo inference. 'MC' denotes the accuracy computed by Monte Carlo procedure, and 'ASJ' is representing the self-joint models that apply auxiliary data during training.

| Model | CIFAR-10 | CIFAR-100 | STL-10 | SVHN |
|---|---|---|---|---|
| Dropout | 95.88±0.04 | 79.77±0.01 | 78.58±0.83 | 97.55±0.01 |
| MC-Dropout | 95.75±0.04 | 79.73±0.07 | 78.42±0.97 | 97.53±0.03 |
| SJ (ours) | 97.03±0.04 | 81.16±0.03 | **87.69±0.51** | 97.59±0.1 |
| MC-SJ (ours) | 96.73±0.02 | **82.55±0.1** | 85.94±2.31 | **97.88±0.08** |
| ASJ (ours) | **97.3±0.0** | 81.39±0.08 | **87.18±0.0** | 97.54±0.0 |
| MC-ASJ (ours) | 95.78±0.15 | 81.16±0.54 | 82.18±2.73 | 97.46±0.0 |

Table 6: Classification accuracy of stochastic models with different sample sizes. Larger sample size results in higher accuracy for self-joint models.

| Model | Sample Size | CIFAR-10 | CIFAR-100 | STL-10 | SVHN |
|---|---|---|---|---|---|
| MC-Dropout | 20 | 95.69±0.07 | 79.68±0.16 | 78.33±1.09 | 97.54±0.03 |
| MC-Dropout | 40 | 95.79±0.07 | 79.63±0.01 | 78.33±0.96 | 97.53±0.05 |
| MC-Dropout | 50 | 95.75±0.04 | 79.73±0.07 | 78.42±0.97 | 97.53±0.03 |
| MC-SJ (ours) | 20 | 96.58±0.16 | 82.03±0.16 | 85.21±2.63 | 97.79±0.02 |
| MC-SJ (ours) | 40 | 96.66±0.04 | 82.32±0.03 | 85.51±2.38 | 97.87±0.02 |
| MC-SJ (ours) | 50 | 96.73±0.02 | 82.55±0.1 | 85.94±2.31 | 97.88±0.08 |
| MC-ASJ (ours) | 20 | 94.22±0.04 | 78.81±0.05 | 75.92±4.75 | 97.45±0.05 |
| MC-ASJ (ours) | 40 | 95.51±0.01 | 80.44±0.39 | 80.41±2.66 | 97.46±0.03 |
| MC-ASJ (ours) | 50 | 95.78±0.15 | 81.16±0.54 | 82.18±2.73 | 97.46±0.0 |

Furthermore, to limit computational cost, in this experiment we restrict sample pairs to $X = X'$. The results in Table 7 highlight that self-joint models are successfully learning conditional independency relations. Moreover, this table provides clear evidence that the independence relations are learned more precisely for in-distribution data than other unfamiliar datasets.

### A.5.4 ADVERSARIAL ROBUSTNESS EXPERIMENTS

In the main body of the paper we reported adversarial robustness against PGD attack with two different setups. This section shows additional adversarial robustness experiments. First, we include an additional setup for PGD, where the maximum perturbation is limited to 3 and the maximum iterations is set to 4. Following the same setting as the main paper, this experiment applies 20 steps for finding the adversarial examples using expectation over transformation (Athalye et al., 2018). Similarly, to produce robust predictions, 20 iterations of Monte-Carlo sampling is applied. We also repeat these experiments with different perturbations ($\epsilon = 2, 3, 4$) for 40 steps of expectation over transformation that follows 40 iterations of Monte-Carlo sampling for defence. Finally, we include

Table 7: Independence error for different self-joint models trained on CIFAR-10/100, SVHN, and STL-10. Low independence errors indicate that self-joint models could learn the independence relation successfully. In addition, lower errors on the trained distribution than other datasets showcase the advantage of learning independency relations over assuming it.

| Model | test dataset | | | |
|---|---|---|---|---|
| | SVHN | STL10 | CIFAR-10 | CIFAR-100 |
| SVHN | 0.0011±0.0 | 0.0072±0.0003 | 0.0074±0.0004 | 0.0074±0.0004 |
| STL-10 | 0.007±0.0034 | 0.0044±0.001 | 0.0083±0.0017 | 0.0075±0.0021 |
| CIFAR-10 | 0.0076±0.0006 | 0.0049±0.0001 | 0.0023±0.0001 | 0.0084±0.0 |
| CIFAR-100 | 0.0001±0.0 | 0.0001±0.0 | 0.0001±0.0 | 0.0±0.0 |

Table 8: Adversarial robustness of different models measured by the classification accuracy(% ± *stdev*) against adversarial examples generated by PGD for different levels of perturbation.

| Model | CIFAR-10 | CIFAR-100 | STL-10 | SVHN |
|---|---|---|---|---|
| Standard ($\epsilon = 2/256$) | 33.2± 4.86 | 9.52± 1.83 | 21.22± 3.53 | 63.04± 3.43 |
| Standard ($\epsilon = 3/256$) | 25.92± 4.77 | 7.23± 2.08 | 15.8± 2.41 | 50.37± 3.41 |
| Standard ($\epsilon = 4/256$) | 22.06± 3.8 | 5.85± 1.76 | 14.0± 2.14 | 46.33± 2.72 |
| $\epsilon = 2/256$, MC sample size=20, EOT steps=20 | | | | |
| Dropout | 44.91± 1.55 | 14.6± 2.16 | 34.1± 3.68 | 67.47± 1.94 |
| SJ (ours) | 62.29± 4.73 | 32.65± 2.09 | 48.12± 5.98 | 82.0± 1.93 |
| $\epsilon = 3/256$, MC sample size=20, EOT steps=20 | | | | |
| Dropout | 38.04± 1.84 | 9.02± 0.98 | 24.98± 3.13 | 54.78± 2.76 |
| SJ (ours) | 52.23± 3.09 | 23.6± 2.08 | 38.42± 6.15 | 74.83± 1.54 |
| $\epsilon = 4/256$, MC sample size=20, EOT steps=20 | | | | |
| Dropout | 33.42± 2.28 | 5.25± 0.82 | 18.08± 1.87 | 46.08± 4.1 |
| SJ (ours) | 45.52± 4.17 | 20.33± 2.83 | 35.29± 6.19 | 70.85± 2.17 |
| $\epsilon = 2/256$, MC sample size=40, EOT steps=40 | | | | |
| Dropout | 45.06± 2.58 | 14.46± 1.48 | 34.63± 1.79 | 67.02± 2.36 |
| SJ (ours) | 58.4± 3.83 | 29.42± 2.04 | 44.27± 4.35 | 80.73± 2.41 |
| $\epsilon = 3/256$, MC sample size=40, EOT steps=40 | | | | |
| Dropout | 36.58± 1.42 | 7.83± 1.2 | 23.96± 1.55 | 54.69± 1.86 |
| SJ (ours) | 47.62± 4.12 | 22.29± 1.71 | 34.23± 5.69 | 75.12± 1.27 |
| $\epsilon = 4/256$, MC sample size=40, EOT steps=40 | | | | |
| Dropout | 31.58± 1.9 | 5.63± 1.02 | 18.52± 2.59 | 47.63± 3.77 |
| SJ (ours) | 41.85± 2.36 | 17.5± 1.58 | 32.42± 5.37 | 69.54± 1.3 |
| $\epsilon = 2/256$, MC sample size=40, EOT steps=20 | | | | |
| Dropout | 45.67± 2.02 | 14.98± 1.68 | 35.17± 3.17 | 66.85± 2.02 |
| SJ (ours) | 61.54± 3.4 | 32.79± 1.35 | 47.98± 6.18 | 83.1± 1.87 |
| $\epsilon = 3/256$, MC sample size=40, EOT steps=20 | | | | |
| Dropout | 38.4± 2.24 | 8.21± 0.98 | 23.98± 3.26 | 55.35± 2.86 |
| SJ (ours) | 49.9± 3.65 | 25.4± 1.95 | 39.37± 7.02 | 75.94± 2.18 |
| $\epsilon = 4/256$, MC sample size=40, EOT steps=20 | | | | |
| Dropout | 32.52± 3.16 | 6.0± 1.16 | 18.02± 1.82 | 47.19± 3.69 |
| SJ (ours) | 44.75± 2.66 | 22.19± 1.96 | 35.75± 6.2 | 72.83± 1.55 |

a setting, where adversarial examples are generated by 20 steps of expectation over transformation, whereas the robust inference applies 40 iterations of Monte-Carlo sampling.

Table 8 shows the performance of different models (two models per method) on 500 adversarial examples that are generated from random test samples (five repetitions per model) for these settings. Note that the step size of PGD is always set to $1/256$ in our adversarial experiments. These results reiterate that the stochastic self-joint model outperforms the rest of models for different levels of perturbation on every dataset. It is also evident that both MC-dropout and self-joint models generally perform better with larger (Monte-Carlo) sample sizes when attack strength is unchanged. Nevertheless, this improvement is more remarkable for self-joint models.

Table 9: AUROC (%) for different models trained on CIFAR-10/100, SVHN, and STL-10. Note that each Outlier Exposure model is explicitly trained on a OOD dataset indicated by (†) as OOD data. Each ASJ model is also utilizing the dataset highlighted by (†) as auxiliary data.

| Model | OOD dataset | | | | | |
|---|---|---|---|---|---|---|
| | SVHN | STL10 | CIFAR-10 | CIFAR-100 | CelebA | VOC |
| **CIFAR-10** | | | | | | |
| Standard | 90.29±3.13 | 64.35±2.15 | | 86.79±0.92 | 76.82±4.48 | 79.05±1.44 |
| Dropout | 93.21±1.37 | 65.94±2.26 | | 89.77±0.96 | 80.4±1.02 | 81.45±1.49 |
| OE | 98.78±0.57 | 68.59±1.71 | | †97.65±0.63 | 98.39±0.97 | 87.76±1.03 |
| SJ (ours) | 93.38±0.84 | 67.25±1.43 | | 93.16±0.59 | 86.69±1.25 | 86.57±0.93 |
| ASJ (ours) | 98.71±0.27 | 70.92±1.52 | | †98.5±0.24 | 99.65±0.19 | 88.92±1.26 |
| **CIFAR-100** | | | | | | |
| Standard | 81.7±1.15 | 78.58±1.15 | 78.44±1.03 | | 58.88±2.16 | 76.62±1.16 |
| Dropout | 80.95±1.86 | 81.65±1.11 | 80.23±1.4 | | 57.95±3.42 | 76.97±1.42 |
| OE | 77.38±1.23 | 94.78±0.43 | †98.01±0.43 | | 54.54±2.99 | 81.48±1.04 |
| SJ (ours) | 80.76±1.22 | 82.64±0.47 | 81.24±0.91 | | 60.86±2.32 | 79.67±1.47 |
| ASJ (ours) | 79.02±2.09 | 94.66±0.93 | †96.42±0.66 | | 53.42±2.26 | 82.91±1.01 |
| **STL-10** | | | | | | |
| Standard | 70.48±3.72 | | 68.87±1.45 | 68.32±2.27 | 67.54±2.23 | 69.63±1.08 |
| Dropout | 69.37±1.59 | | 71.31±2.08 | 67.96±0.63 | 73.3±1.28 | 72.41±1.77 |
| OE | 78.74±2.79 | | 66.45±2.87 | 70.56±1.75 | 71.1±1.88 | 66.84±1.45 |
| SJ (ours) | 84.17±7.85 | | 82.66±3.62 | 80.24±4.22 | 83.92±2.33 | 80.33±1.68 |
| ASJ (ours) | 86.16±7.51 | | 83.25±2.32 | 84.62±2.92 | 88.73±1.71 | 81.4±0.97 |
| **SVHN** | | | | | | |
| Standard | | 96.26±1.38 | 95.48±1.21 | 95.31±1.44 | 97.2±0.71 | 96.2±1.3 |
| Dropout | | 96.83±0.87 | 96.56±0.94 | 96.46±0.67 | 97.65±0.37 | 96.59±0.8 |
| OE | | 99.96±0.06 | 100.0±0.0 | †100.0±0.01 | 99.98±0.05 | 100.0±0.01 |
| SJ (ours) | | 99.03±0.16 | 98.88±0.38 | 98.65±0.19 | 99.23±0.31 | 98.77±0.22 |
| ASJ (ours) | | 99.99±0.01 | 99.97±0.04 | †99.98±0.06 | 100.0±0.0 | 99.99±0.01 |

### A.5.5 DETECTING OOD DATA

Robustness against OOD data is an important aspect of stochastic self-joint models. Here, we extend the results in the main body of the paper for each OOD dataset. The datasets that we consider for OOD include: SVHN, STL-10, CIFAR-10, CIFAR-100, CelebA, and VOC. All these datasets contain natural color images. Note that we did not remove any overlapping class from these datasets. However, the majority of these datasets have little to no common classes, except for STL-10 and CIFAR-10, which share $80\%$ of their classes. We report the area under the receiver operating characteristic curve, *AUROC*, and the *detection error* for each model against each OOD dataset.

We randomly select $500$ samples from test set of in-distribution and OOD data to compare the performance of each model (five repeats). Maximum softmax probability (e.g. Hendrycks & Gimpel (2016)) is applied for detection score. However, we apply entropy for Monte-Carlo dropout models, because we found it slightly superior for these models. Finally, the Monte-Carlo approximation always compute average of $50$ samples.

Tables 9 and 10 show the performance of different methods on OOD detection in AUROC and detection errors metrics, respectively. The tables suggest that outlier exposure (OE) (Hendrycks et al., 2018) and self-joint models trained with auxiliary data perform almost always better than the rest. These results suggest learning to make predictions independent of OOD data can effectively reduce vulnerability to unseen OOD samples.

Table 10: Detection Error(%) for different models trained on CIFAR-10/100, SVHN, and STL-10. Note that each Outlier Exposure model is explicitly trained on a OOD dataset indicated by (†) as OOD data. Each ASJ model is also utilizing the dataset highlighted by (†) as auxiliary data.

| | OOD dataset | | | | | |
|---|---|---|---|---|---|---|
| Model | SVHN | STL10 | CIFAR-10 | CIFAR-100 | CelebA | VOC |
| **CIFAR-10** | | | | | | |
| Standard | 14.67±3.5 | 36.03±1.93 | | 17.78±0.94 | 27.98±3.59 | 24.57±1.12 |
| Dropout | 12.75±1.38 | 35.54±2.21 | | 16.01±0.86 | 23.5±0.8 | 22.52±1.4 |
| OE | 3.39±0.54 | 34.65±1.7 | | †5.87±0.76 | 2.85±0.55 | 17.93±0.98 |
| SJ (ours) | 11.91±1.15 | 34.92±0.92 | | 12.54±0.77 | 18.54±2.12 | 19.01±0.77 |
| ASJ (ours) | 4.94±0.55 | 33.46±1.31 | | †5.06±0.44 | 2.06±0.3 | 15.77±1.44 |
| **CIFAR-100** | | | | | | |
| Standard | 24.02±0.9 | 26.5±0.92 | 26.71±1.13 | | 39.37±1.25 | 28.46±1.18 |
| Dropout | 28.04±1.13 | 25.12±0.91 | 26.49±1.18 | | 39.31±2.33 | 28.78±1.48 |
| OE | 24.63±0.89 | 11.51±0.46 | †5.9±0.87 | | 40.64±2.54 | 25.88±0.79 |
| SJ (ours) | 25.51±0.87 | 24.1±0.77 | 25.49±0.97 | | 37.06±1.61 | 26.37±1.2 |
| ASJ (ours) | 25.59±1.27 | 10.9±0.8 | †8.3±1.2 | | 40.33±2.02 | 23.46±1.0 |
| **STL-10** | | | | | | |
| Standard | 32.56±3.17 | | 34.24±1.09 | 33.91±1.84 | 35.21±1.77 | 33.61±0.93 |
| Dropout | 33.77±1.89 | | 32.35±1.19 | 34.74±0.85 | 29.78±1.21 | 32.02±1.31 |
| OE | 24.78±3.9 | | 36.49±2.25 | 33.4±0.95 | 32.19±0.75 | 35.88±0.92 |
| SJ (ours) | 22.39±7.26 | | 23.6±3.39 | 25.58±3.25 | 20.72±1.9 | 25.71±1.76 |
| ASJ (ours) | 17.94±4.06 | | 21.86±1.57 | 21.24±2.4 | 16.69±1.86 | 24.73±1.3 |
| **SVHN** | | | | | | |
| Standard | | 7.86±1.35 | 8.6±0.98 | 8.82±1.5 | 6.41±0.78 | 7.77±1.68 |
| Dropout | | 7.21±1.04 | 7.38±0.91 | 7.18±0.81 | 5.77±0.64 | 7.25±0.78 |
| OE | | 0.06±0.05 | 0.0±0.0 | †0.05±0.05 | 0.03±0.07 | 0.03±0.05 |
| SJ (ours) | | 3.69±0.59 | 3.85±0.67 | 4.81±0.61 | 3.05±0.6 | 4.32±0.46 |
| ASJ (ours) | | 0.07±0.08 | 0.2±0.12 | †0.19±0.15 | 0.06±0.07 | 0.13±0.09 |

