# OpenReview forum: "Self-Joint Supervised Learning"
_ICLR.cc/2022/Conference — ICLR 2022 Poster_

### Official Review · Reviewer_sxB6 · 2021-11-02

**Correctness:** 4
**Technical Novelty And Significance:** 2
**Empirical Novelty And Significance:** 4
**Recommendation:** 8
**Confidence:** 4

**Main Review:**

The empirical results look convincing, and the Authors' interpretation of the method as regularising training sounds convincing. The term Y_i Y_j^T in Eq (9) means that maximising the MLE (8) fully would lead to a classifier which evaluates each sample independently. Hence, classifying two samples at a time matters only if MLE is not fully optimised. Hence, it's a regularisation scheme, as the Authors say. If the model architecture does not ensure the factorisation of the h() function by design, the model has to competing objectives to fulfill: get better at classifying samples correctly and get better at classifying them independently. Also, a training dataset containing N examples produces N^2 pairs of examples, which makes the training loop harder for the model. This means it's harder to overfit it.

However, if the model architecture was big enough, it would probably overfit nevertheless, because it would be able to produce 100% conditionally independent label predictions and overfit each of them independently. This raises a question: could a similar effect could be achieved simply by training a smaller model, or stopping training earlier?

Some parts of the paper are not clear:
- In the Introduction, the Authors say "By training with a pair instead of one sample, the model explicitly learns sample-to-sample relationships". But if the labels are deterministic (as is most often the case), there are no such relationships (deterministic random variables are always independent). What "relationships" are learnt then?
- When computing the gradient for training, are the pseudo-labels \hat{Y} differentiated too?
- How were the network architectures modified to support increased input and output dimensionality. Are the images concatenated together? The output layer must be larger (c^2 instead c logits), but what about the hidden layers?
- How is the model capacity compared between standard and joint models in Figure 2? What does x = 10 mean on the X axis of the plot? Do both models have the same number of weights? Or the joint one has 2x as many?

Finally, there are many small problems with writing and style, which should be corrected (e.g. " using loss penalties and to use stochas-
ticity", " Even a well-trained model could benefit from having additional data available at the time of inference much, as a human expert can still benefit from having", "Although, these approaches learn...", etc.)

**Summary Of The Paper:**

The Authors propose a new way of training classifier models. Instead of classifying each i.i.d. example independently during training and inference, they jointly classify a pair (X1, X2) of them, returning a joint distribution of labels (Y1, Y2). The loss function is modified so that the model learns that labels Y1 and Y2 are conditionally independent given X1 and X2.

This approach is shown experimentally to lead to superior model performance for the same model capacity (measured as relative layer width). The Authors attribute it to the regularising effect of new training method. Additionally, joint training on pairs of examples allows the use of auxiliary unlabelled examples.


**Summary Of The Review:**

The presented method is interesting and should be accepted for publication. However, the paper could be improved and there are some open questions which should be investigated.

---

### Official Review · Reviewer_Ky9n · 2021-11-03

**Correctness:** 3
**Technical Novelty And Significance:** 2
**Empirical Novelty And Significance:** 3
**Recommendation:** 5
**Confidence:** 3

**Main Review:**

Strength:

1. The main idea of modelling the conditional joint label distribution of a pair of data samples of the paper is straightforward. The proposal of the parameterisation of the conditional joint label distribution and strategies for making predictions in the inference phase is intuitive.

2. The experiments cover a wide range of applications.

3. The paper is well written and easy to follow.

Weakness:

1. I think it is analytically unjustified how or why explicitly modelling conditional independence can help deal with overfitting, adversarial examples, detecting OOD data, although the paper shows a considerable amount of experiments in various applications. For example, it is not clear where the robustness against adversarial attacks of the proposed method comes from. I would expect some theoretical or analytical study on this point. Also on this point, in Eq. (9), $Y_iY^T_j$ indicates the true labels of two samples, which is a one-hot matrix, without any label correlation. It is hard for me to understand why MLE in this case helps.

2. Although the paper studies several applications, it is a bit hard to position the paper in the literature. Specifically, for each specific applicaltion, the paper mainly compares with the fundamental methods. But there is a rich literature in each application. For example, in adversarial machine leanring, there are many advanced methods that improve adversarial robustness without adversarial training, e.g. in [1] where an ensemble model is used. I'm wondering how the proposed method performs in the comparion with the state-of-the-art methods in their specific application.

[1] Pang, Tianyu, et al. "Improving adversarial robustness via promoting ensemble diversity." International Conference on Machine Learning. PMLR, 2019.

3. It would be interesting to study the complexity (running speed) of the proposed method in both training and inference phases. As the proposed might need to sample multiple pairs of data to make prediction of one data sample, it is important to show how much addtional complexity would be added in the inference phase.


**Summary Of The Paper:**

This paper introduces a pairwise loss function for supervised learning, named self-joint learning. Instead of predicting the conditional distribution of the label given one data sample as in conventional supervised learning, the proposed self-joint learning framework predicts the conditional joint label distribution of a pair of data samples. The paper proposes the corresponding constructions of the conditional joint label distribution and several strategies for making predictions in the inference phase. The paper shows empirical studies on dealing with overfitting, adversarial robustness, and detecting OOD data.

**Summary Of The Review:**

The paper studies an interesting idea and shows that the proposed idea improves some baselines in several applications. But I feel that it needs additional study on why the idea works and more comparison with more advanced methods.

---

> ### Comment · Reviewer_sxB6 · 2021-11-18
> **Comments on the review**
>
> Reviewer Ky9n,
>
> The Authors did not send any response, unfortunately, but that doesn't mean the reviewers cannot have a discussion between themselves :)
>
> I wanted to briefly comment about the weaknesses you have pointed out.
>
> 1. Lack of theoretical analysis: I agree it would be better to have it, but the empirical results are strong enough IMHO to warrant publication on their own. Theoretical analysis can be tackled later. A paper doesn't have to solve a problem completely.
>
> 2. Comparison with specialised methods: again, it would be better to have it, but I think general methods also have their place, for example when dealing with multi-modal inputs. And it's perhaps more fair to compare a general method with other general methods.
>
> 3. Completely agree on that point, running time should be reported.

---

### Official Review · Reviewer_PZCK · 2021-11-03

**Correctness:** 4
**Technical Novelty And Significance:** 4
**Empirical Novelty And Significance:** 4
**Recommendation:** 8
**Confidence:** 5

**Main Review:**

Strengths:
1. Simple idea that works very well
2. Paper is easy to understand
3. Formulation of the problem is well grounded theoretically
4. Shows advantages over several different areas of deep learning

Areas of improvements:
1. While results are good, there is a lack of explanation on why the results are good. My sense is that the combinations of pairs of data span a much larger input space than to feed data one by one into the network. In a sense the amount of ‘independent’ data grows exponentially. This is a huge regulariser
2. The data is not strictly independent since the authors feed different combinations of pairs. In any case, iid assumptions is not strictly enforce in many real world problems
3. Icing on the cake would be having some theoretical proofs of properties of this network. Especially some regularising properties.

Queries:
1. How is the images being fed in? Side-by-side or concat channel wise? Fig1 seems to suggest side-by-side.

**Summary Of The Paper:**

Paper propose a simple but effective method of learning. Feeding in a pair of data and learn the join conditional probabilities for predictions. The advantage is that the combination of data goes as the number of ways to partition the original data into pairs. This is a huge number.


**Summary Of The Review:**

Good piece of work, can be enhance some more with more extensive analysis of the results. E.g. more explanations of the observations.

---

### Decision · Program_Chairs · 2022-01-20

**Decision:**

Accept (Poster)

**Comment:**

This paper explores a classification approach based on labeling pairs of inputs concurrently using a single network, rather than singletons. The authors test the approach on adversarial robustness (towards norm bounded perturbations), OOD detection next to basic standard accuracy calculations.

While the key idea is potentially interesting and the paper has received positive comments from the majority of reviewers, there were also some concerns that need to be addressed in a final manuscript:

* The paper does not motivate or explain theoretically why the joint classification framework is superior, beyond verbose arguments. These
arguments need to be better clarified and linked with the experimental evaluation.

* While the empirical results are perceived as positive by the reviewers, one reviewer has raised the concern about the comparisons. The adversarial robustness and OOD comparisons are indeed basic. The adversarial attack used here is quite a weak PGD attack with a small radius and low iteration budget. Possibly include stronger attacks. The OOD comparisons are with standard baselines only. Please include further comparisons.

In its current form, the paper seems to be acceptable, and I strongly encourage the authors to improve both the theoretical justification, and empirical exploration in the final version.